# Nonalcoholic Fatty Liver Disease and Endocrine Axes—A Scoping Review

**DOI:** 10.3390/metabo12040298

**Published:** 2022-03-29

**Authors:** Madalena Von-Hafe, Marta Borges-Canha, Catarina Vale, Ana Rita Leite, João Sérgio Neves, Davide Carvalho, Adelino Leite-Moreira

**Affiliations:** 1Departamento de Cirurgia e Fisiologia, Faculdade de Medicina da Universidade do Porto, 4200-319 Porto, Portugal; madalenavonhafe@gmail.com (M.V.-H.); catarinavale9@hotmail.com (C.V.); ana.leite7c@gmail.com (A.R.L.); joaosergioneves@gmail.com (J.S.N.); amoreira@med.up.pt (A.L.-M.); 2Serviço de Endocrinologia, Diabetes e Metabolismo do Centro Hospitalar Universitário de São João, Alameda Prof. Hernâni Monteiro, 4200-319 Porto, Portugal; davideccarvalho@gmail.com; 3Investigação e Inovação em Saúde (i3s), Faculdade de Medicina da Universidade do Porto, 4200-319 Porto, Portugal; 4Serviço de Cirurgia Cardiotorácica do Centro Hospitalar Universitário de São João, 4200-319 Porto, Portugal

**Keywords:** NAFLD, fatty liver, lipidomics, metabolic modeling, endocrine disorders

## Abstract

Nonalcoholic fatty liver disease (NAFLD) is the leading cause of chronic liver disease. NAFLD often occurs associated with endocrinopathies. Evidence suggests that endocrine dysfunction may play an important role in NAFLD development, progression, and severity. Our work aimed to explore and summarize the crosstalk between the liver and different endocrine organs, their hormones, and dysfunctions. For instance, our results show that hyperprolactinemia, hypercortisolemia, and polycystic ovary syndrome seem to worsen NAFLD’s pathway. Hypothyroidism and low growth hormone levels also may contribute to NAFLD’s progression, and a bidirectional association between hypercortisolism and hypogonadism and the NAFLD pathway looks likely, given the current evidence. Therefore, we concluded that it appears likely that there is a link between several endocrine disorders and NAFLD other than the typically known type 2 diabetes mellitus and metabolic syndrome (MS). Nevertheless, there is controversial and insufficient evidence in this area of knowledge.

## 1. Introduction

Nonalcoholic fatty liver disease (NAFLD) is the leading cause of chronic liver disease. NAFLD is a metabolic liver disease that encompasses a wide spectrum from simple steatosis to steatohepatitis (NASH) and fibrosis to cirrhosis and hepatocarcinoma [1]. It has also been termed a “barometer of metabolic health” due to its metabolic roots [2]. A group of experts on the theme have recently reached a consensus, saying that this entity might be known as MAFLD (metabolic-associated fatty liver disease) and be diagnosed by positive criteria rather than being an exclusion diagnosis (requiring the exclusion of other causes chronic liver diseases before diagnosing NAFLD) [3]. Its high prevalence, complex pathogenesis, and lack of approved therapies make this disease a hot topic of scientific research [4].

NAFLD often occurs associated with endocrinopathies, as it has been increasingly recognized [5,6]. Digging into these associations may improve the current knowledge on this disease [7]. Moreover, emerging evidence suggests that endocrine dysfunction may play an important role in NAFLD development, progression, and severity [8]. Moreover, a variety of rare hereditary liver and intestinal diseases as well as several drugs may trigger or worsen NAFLD; this further highlights the complexity in understanding NAFLD [9].

In this work we aimed to summarize and clarify the data published on the crosstalk between NAFLD and different endocrine axes. This is an exploratory review that addresses a broad question and does not intend to be a systematic review; this is the reason why we did not follow PRISMA guidelines. This review does not include associations between NAFLD and diabetes mellitus, obesity, metabolic syndrome, or adipose or pancreatic tissues (as well as their related hormones) since those have already been extensively studied and described in the literature [7]. Here, we present our results, organized by endocrine axis.

## 2. Hypothalamic and Pituitary Dysfunction

### 2.1. Growth Hormone (GH)

#### 2.1.1. Adult GH Deficiency

While essential for linear growth during childhood, GH promotes lipolysis, especially in visceral adipose tissue and protein synthesis and decreases peripheral insulin sensitivity and glucose uptake in adults [10]. Adult GH deficiency is a clinical syndrome that mainly results from pituitary tumors or the treatment of such tumors, namely, with surgery or radiation. Regardless of etiology, GH deficiency is generally associated with several metabolic changes, including increased visceral adipose tissue, decreased lean body mass, and dyslipidemia, with higher total and low-density lipoprotein (LDL) cholesterol, triglycerides, and hypertension [11,12]. Although GH has antagonistic effects on insulin action, its deficiency leads to insulin resistance and glucose intolerance, probably due to specific changes in fat distribution, such as increased visceral adipose tissue and ectopic fat accumulation [10,13]. These frequently lead to metabolic syndrome (MS) in patients with untreated GH deficiency [13]. Given the intricate association between MS and NAFLD, a focus has been given to the role of GH in the pathogenesis of NAFLD in the last decade (Figure 1).

Several cross-sectional studies reported an increased prevalence of liver dysfunction and NAFLD in patients with hypopituitarism, particularly those with GH deficiency [14,15,16]. Moreover, patients with GH deficiency and NAFLD have an accelerated progression of the hepatic disease [15]. Additionally, a study by Koehler et al. showed that obese patients with NASH and advanced fibrosis have low serum GH levels and that normal GH levels essentially excluded advanced fibrosis [17,18]. Patients with Laron syndrome, which is characterized by GH resistance due to inactivating mutations of the GH receptor, were also found to have a higher incidence of NAFLD [19]. Interestingly, acromegalic patients treated with the GHR antagonist pegvisomant showed increases in hepatic triglyceride content [20]. Despite the aforementioned results, evidence on this matter remains conflicting, with some studies not finding differences in the prevalence of NAFLD or in the intrahepatic lipid content assessed by magnetic resonance spectroscopy between GH deficient patients and healthy adults [21]. Heterogeneity in sample sizes and the clinical characteristics of included patients, such as sex and ethnicity, may help to explain these differences in the studies’ results. 

Animal and cell culture studies further support a role for the GH/ insulin-like growth factor-1 (IGF-1) axis in the pathophysiology of NAFLD and its progression to NASH and fibrosis [11]. Animal models with liver-specific mutations in the GH receptor or downstream signaling pathways (JAK2/STAT5) develop metabolic syndrome, hepatic steatosis, steatohepatitis, and fibrosis [22,23,24]. Moreover, in animal models of adult-onset hepatic GH resistance, steatosis and NASH evolve rapidly after the loss of hepatic GH signaling, regardless of other signs of metabolic dysfunction [25]. 

The restoration of GH levels in adults with GH deficiency reduces body fat, increases lean mass, and ameliorates the lipid profile, decreasing total and LDL cholesterol and increasing high-density lipoprotein (HDL) cholesterol [26,27]. A few small studies have shown that GH replacement improves hepatic injury, as observed by a rapid decrease in serum liver transaminases and gamma-glutamyl transferase levels, steatosis, lobular inflammation, hepatocyte ballooning, and the severity of fibrosis [15,28]. Additionally, GH supplementation in pediatric patients with GH deficiency has been associated with an improvement in NAFLD, as it decreases visceral fat accumulation and lipid deposition on the liver and enhances mitochondrial function [29,30,31]. Animal studies have reinforced these results [24]. It is well known that the actions of GH are mediated both directly and indirectly through the stimulation of IGF-1 production [10]., and the majority of circulating IGF-1 (>90%) is produced by hepatocytes in response to growth hormone receptor stimulation [32]. Increasing evidence suggests that both GH and IGF-1 have direct and indirect effects on hepatic structure and function [33]. Moreover, decreased GH and, consequently, IGF-1 might be responsible for the muscle mass changes, particularly sarcopenia, which is seen in NAFLD [34]. GH was recently proposed to directly inhibit de novo lipogenesis and the expression of peroxisome proliferator-activated receptor-gamma (PPAR-γ) and CD36, key regulators of free fatty acid uptake [23,25]. IGF-1 is an anti-inflammatory molecule that contributes to mitochondrial function and reduces oxidative stress in the liver [24]. IGF-1 prevents cholesterol accumulation through the stimulation of the expression of ATP-binding cassette transporter A1 (ABCA1), a pivotal regulator of lipid efflux from cells to apolipoproteins [35]. Furthermore, IGF-1 limits the activity of hepatic stellate cells and induces their senescence, therefore attenuating hepatic fibrosis [36]. Lower levels of IGF-1 might result in lower protection against liver inflammation and fibrosis [37]. By impairing the adipose tissue phenotype, GH deficiency increases the expression of proinflammatory cytokines and adipokines [38], which compromises insulin sensitivity and impairs the ability of adipose tissue to store fat, increasing lipid influx into ectopic organs, such as the liver [22]. Chronic liver diseases, including NASH, are associated with a reduction of GH receptor (GHR) expression and, therefore, reduced IGF-1 levels [39]. As described above, lower levels of IGF-1 impair liver homeostasis, with a higher risk of fibrosis, leading to a vicious cycle of both worsening hepatic homeostasis and increasing growth hormone function.

Given the crucial role of GH in hepatic lipid metabolism, there are some clinical trials examining the impact of low-dose GH supplementation in patients with hepatic steatosis and NASH without known hypothalamic/pituitary disease. A new clinical trial had its results recently published, showing that treatment with recombinant human GH may have the potential to reduce liver fat content in adolescents with NAFLD and obesity [40]. Other clinical trials studying the impact of GH supplementation on NAFLD are underway, such as the clinical trial named Growth Hormone and Intrahepatic Lipid Content in Patients With Nonalcoholic Fatty Liver Disease (NCT02217345). Lastly, IGF-1 replacement is also being considered as an option to treat patients with liver diseases [41]. Experimental studies show that treatment with IGF-1 is particularly beneficial in the reduction of liver fibrosis, although positive effects in hepatic steatosis and inflammation can also be seen [24,42]. 

#### 2.1.2. Acromegaly

Acromegaly is characterized by excessive GH and, consequently, IGF-1 and is the most frequent cause of GH-secreting pituitary adenoma. Increased levels of GH are associated with increased lipolysis and favorable body composition, with increased lean body mass and decreased visceral and subcutaneous adipose tissue [43]. However, acromegaly also promotes insulin resistance, with consequent hyperglycemia, hyperinsulinemia, hypertriglyceridemia, and an increased risk of overt diabetes [44]. These paradoxical effects may justify contradictory evidence on this topic. While some studies including patients with active acromegaly found that intrahepatic lipid, measured by magnetic resonance spectroscopy, is relatively low in comparison to healthy subjects [20,45], others showed that hepatic steatosis is a common comorbidity in acromegaly, hypothesizing that lipotoxicity and insulin resistance may outweigh the direct hepatic effects of GH [46].

Acromegaly treatment with surgery or medical therapy improves metabolic risk by increasing insulin sensitivity [45]. However, GH, IGF-1, insulin-like growth factor binding proteins (IGFBPs), and medical treatment have a complex relationship with insulin sensitivity and hepatic steatosis. GHR antagonists (GHRA) induce an improvement in acromegaly glycemic control through the decrease of glucose and the normalization of insulin secretion [47]. This effect enables one to understand the important effect of GH on hepatic and peripheral IGF-1 action. Hepatic GH-induced IGF-1 production is regulated by portal insulin levels, as insulin promotes the translocation of hepatic GHR to the surface. When portal insulin levels are high, the liver becomes GH sensitive, regardless of the cause of the increased production of insulin. In addition, portal insulin also inhibits hepatic IGFBP1 production, which may increase the bioavailability of circulating IGF-1. Insulin suppression by somatostatin analogs (SSA) also selectively results in hepatic GH resistance, which itself decreases hepatic IGF-1 production [48]. Therefore, the consequent reduction in circulating IGF-1 does not necessarily reflect GH activity in peripheral tissues. It, thus, makes sense that the normalization of serum IGF-1 levels during SSA does not necessarily imply the control of disease’s activity in peripheral tissues, which is a condition that Neggers coined as being “extra-hepatic acromegaly” [49]. This concept received support in a study that evaluated surgically and SSA-treated acromegalics. Despite the normalization of IGF-1, SSA-treated patients had less suppressed GH levels and less symptom relief [50]. On the other hand, GHRA do not block all tissues with equal effectiveness for the GH actions. Adipose tissue seems to require less GHRA to reduce GH actions when compared to the liver, where more GHRA are required to reduce IGF-1 production [51]. This could be a reason for local GHRA-induced lipomatosis. In further support of this hypothesis, it was recently reported that short-term GHRA administration in healthy subjects can suppress lipolysis without affecting either circulating or local IGF-1 [52]. Accordingly, it is possible that peripheral suppression of GH activity is obtained prior to the normalization of hepatic IGF-1 production. Therefore, GHRA-treated patients with acromegaly and normal peripheral IGF-1 can have peripheral GH deficiency [49]. If all this is true, then patients with acromegaly and diabetes should need higher GHRA doses to normalize IGF-1 compared to patients without diabetes. Recently, this was demonstrated by Droste et al. [53].

In a cross-sectional study including patients previously treated for acromegaly, hepatic steatosis, measured by magnetic resonance spectroscopy, was found to be increased compared to healthy controls, even several years after successful treatment [54]. In a recent prospective study, Ciresi et al. found no differences in the prevalence of steatosis after 12 months of treatment with somatostatin analogs, measured by abdominal ultrasonography [55]. The heterogeneity in the chosen imaging method to assess hepatic steatosis and in the treatments for acromegaly may explain these differences. 

### 2.2. Hyperprolactinemia

Prolactin is a polypeptide hormone produced by lactotroph cells in the anterior pituitary [56]. Prolactin release is mainly controlled by hypothalamic inhibitory tone through dopamine and the stimulatory influences of thyroid stimulating hormone (TSH)-releasing hormone and circulating estrogens. Its major functions are related to pregnancy and lactation [57]. In addition, prolactin is involved in the regulation of the immune system, food intake, and bone formation [56].

A growing body of evidence supports prolactin as an active contributor to human metabolic health [57]. Namely, experimental animal studies found that it stimulates pancreatic β cell proliferation and insulin gene transcription, modulates lipid metabolism in adipose tissue, and induces adipogenesis [58,59]. Despite the beneficial roles of prolactin on metabolic homeostasis, pathological increases in prolactin levels have been frequently associated with metabolic disturbances, namely, weight gain, obesity, hyperinsulinemia, and reduced insulin sensitivity, all considered important players in the pathogenesis of NAFLD [60]. Normalization of prolactin levels with dopamine agonists correlated with weight loss, although some studies have shown a more pronounced weight loss in men, suggesting a gender difference [61]. Furthermore, treatment with dopamine agonists improves insulin sensitivity, glycemic control, and lipid profile, reducing triglycerides and total and LDL cholesterol [62].

The impact of prolactin on liver function and structure is poorly understood (Figure 2). The presence of functional prolactin receptors in hepatocytes has also been previously demonstrated [63]. Recent in vitro and in vivo studies suggest that prolactin protects the liver against lipid accumulation by decreasing the expression of CD36 and stearoyl-coenzyme A desaturase 1 (SCD1), an enzyme involved in fatty acid biosynthesis [64,65]. In rodents, prolactin is thought to be involved in the regulation of liver insulin sensitivity [66]. In human studies, lower prolactin levels were found in patients with more severe hepatic steatosis, suggesting a possible involvement of prolactin in the progression of this disease [64]. Although prolactin is thought to reduce liver fat content, it is plausible that chronic hyperprolactinemia is involved in the development of NAFLD. Despite the absence of studies evaluating liver function and structure in patients with hyperprolactinemia, a few animal studies support this hypothesis. Luque et al. suggest that prolactin may be directly involved in changes in the signaling pathways of de novo lipogenesis, which lead to fatty liver [67]. In diabetic murine models, the triglyceride content in the liver increased with the administration of high doses of prolactin [68]. 

### 2.3. Vasopressin Disturbances

Vasopressin (V), also known as antidiuretic hormone (ADH), has a crucial role in the regulation of water balance, vascular tone, and the endocrine stress response [69]. Due to its short half-life, very low concentration, small size, and poor stability in plasma samples, clinical studies indirectly determine the levels of ADH in the circulation through the measurement of copeptin, which is produced in equimolar quantities with ADH [69]. 

A role of ADH in the regulation of glucose and lipid metabolism has been acknowledged by several epidemiological and experimental studies [70]. Patients with diabetes mellitus have markedly increased levels of ADH in comparison with healthy subjects [71]. Whether copeptin is predictive of the development of diabetes-induced NAFLD remains unknown. Recent studies demonstrated a strong association between high copeptin levels and the prevalence and severity of both NAFLD and NASH [71,72].

Several ex vivo and in vitro studies have shown that ADH intensifies hyperglycemia. It enhances hepatic gluconeogenesis and glycogenolysis through the activation of V1a receptors [73]. ADH also induces vasoconstriction, contributing to liver hypoxia, and further stimulates glycogenolysis [74]. and stimulates the release of pituitary adrenocorticotropin hormone via V1b receptors, increasing the release of cortisol, which is thought to be an important contributor to ADH-induced hyperglycemia and insulin resistance [75]. 

Despite being controversial and less understood, most studies suggest that ADH decreases plasma non-esterified fatty acids [76,77]. ADH promotes lipogenesis in the hepatic tissue through the V1a receptor and inhibits lipolysis in adipocytes [76,78]. The V-induced decrease in plasma non-esterified fatty acids may reduce its supply to the liver [77]. On the other hand, one study found the role of vasopressin in fatty acid synthesis and lipogenesis varied with different incubating media of rat hepatocytes, suggesting that vasopressin function may vary according to the nutritional status [79]. Taveau et al. found reduced levels of hepatic cholesterol and triacylglycerol and a lower expression of genes involved in lipogenesis in well-hydrated obese rats with low levels of ADH [80].

## 3. Phosphocalcic Metabolism Disturbances

### 3.1. Vitamin D Deficiency

Vitamin D is classically recognized for its role in phosphocalcium metabolism and bone health. Notwithstanding, its receptors are present ubiquitously, and it is associated with several effects in various organs and systems [81,82]. Vitamin D deficiency seems to be associated with several metabolic disturbances [83,84,85], namely, NAFLD, although inconsistently.

Fundamental research studies have shown that vitamin D exerts anti-inflammatory, anti-proliferative, and anti-fibrotic effects in the liver [86]. In vitro and in vivo studies suggest that this protective effect may be partially mediated by stellate cells through the inhibition of fibrogenesis [83,86,87] Using mouse hepatocytes, Dong B. et al. showed that vitamin D receptor activation in macrophages by vitamin D ligands ameliorates liver inflammation and steatosis [88]. Moreover, obese animals with vitamin D deficiency presented with greater NAFLD progression, and vitamin D supplementation improved hepatic morphology and function [83]. Contrariwise, Bozic et al. showed that hepatic vitamin D receptor activation promotes high-fat diet-associated liver steatosis in a mouse model [89].

Human studies are also conflicting. Several epidemiological studies lead towards an association between low vitamin D levels and NAFLD, though no causal relationship has been found [90,91,92]. Namely, Targher et al. showed that patients with biopsy-proven NAFLD had lower vitamin D levels compared to controls [93]. Furthermore, an analysis of data from the National Health and Nutrition Examination Survey (NHANES III) claims that vitamin D levels are inversely associated with the severity of NAFLD [94]. A recent study from our group associated vitamin D deficiency with a higher risk of hepatic steatosis in individuals with morbid obesity [95]. An analysis from the Sixth Korea National Health and Nutrition Examination Survey (KNHANES VI) argued against such association [96]. In a Mendelian randomization analysis, Wang et al. found no causal association between vitamin D and NAFLD in a Chinese population with over 9000 participants [97]. Additionally, Barchetta et al. performed a randomized, double-blind, placebo-controlled trial (RCT) and concluded that vitamin D supplementation, in a high oral dose for a period of 24 weeks did not improve hepatic steatosis in patients with NAFLD and type 2 diabetes [98,99,100]. In another RCT in NAFLD patients, no beneficial effects on liver function were seen when comparing supplementation with vitamin D, calcitriol, and placebo [101]. Therefore, no strong recommendations currently exist concerning vitamin D supplementation in patients with NAFLD [102,103].

Pacifico L. et al. dwelled on the possible confounders of a NAFLD/NASH and vitamin D deficiency association, namely, the influence of the host and environment in vitamin D levels as well as the great variability in laboratory methods and intra-individual variability in this vitamin’s level [104]. More studies, particularly well-powered randomized controlled trials, are needed to evaluate the potential role of vitamin D supplementation in the management of NAFLD. 

### 3.2. Other Disturbances of Bone Metabolism

The deterioration of bone homeostasis has been incongruously associated with NAFLD, and the pathophysiology of such association remains to be elucidated [105]. It is hypothesized that very complex mechanisms are involved, and a plausible perspective centralizes the causal pathway in the liver. The dysfunctional visceral adipose tissue that often exists concurrently with NAFLD is a great source of pro-inflammatory, pro-coagulant, and profibrogenic factors that may contribute to this association. The state of insulin resistance commonly present in NAFLD patients may also play an important role as well as the vitamin D deficit that may relate to such hepatic disorders, as previously mentioned [105,106,107].

An observational study in a Chinese population showed that the liver fat content was inversely correlated with bone mineral density (BMD) in middle-aged and elderly men but not in women [108]. Additionally, in a cohort of Chinese participants that were 40 years old or older, the prevalence of osteoporotic fractures was significantly greater in men with NAFLD but not in women [109]. Moreover, a Korean population-based study presented a detrimental effect of NAFLD on BMD in men but an unexpected positive effect in postmenopausal women [110]. There are some studies that show similar harmful effects for both genders (namely postmenopausal women), but others evidence a detrimental effect in postmenopausal women [111,112]. Bhatt et al. stated that PTH levels are independently associated with NAFLD in Asian Indians [113]. Similarly, a metanalysis that included observational studies focusing on bone mineral density claimed that no correlation exists between bone mineral density and NAFLD [114]. It is important to recognise that most of the studies mentioned are in Asian populations, suggesting an ethnic difference in the impact of the phosphocalcic axis on the liver. 

Despite being a controversial theme of the debate, it is of great interest to clarify the existence of such links in future prospective studies, namely, addressing gender, ethnic, and age differences. The routine screening of BMD in NAFLD patients may become an important addition to the management of these complex patients.

## 4. Thyroid Dysfunction

Thyroid dysfunction, explicitly, hypothyroidism, has been proposed as a possible contributory mechanism for the pathophysiology of NAFLD [115] (Figure 3). A large 2018 metanalysis that included a total of 15 studies and 44,140 individuals suggested that hypothyroidism is significantly associated with the presence and severity of NAFLD [116]. It is biologically plausible that the thyroid axis plays an important role in NAFLD development, as thyroid hormones (TH) are crucial in the regulation of numerous metabolic processes, such as cholesterol and lipid metabolism and intra-hepatic concentration, circulating lipoprotein levels, body weight, and insulin resistance [117,118,119,120]. TH regulate the expression of several hepatic lipogenic genes, and recent studies have shown that several genes whose expression are altered in NAFLD are also regulated by TH [121,122]. Nevertheless, there is still controversy on this subject.

At a local level, both human and animal studies show that TH levels are decreased in the livers of individuals with NAFLD, and defective intrahepatic deiodinase expression has recently been proposed as a hallmark of NASH [123]. Namely, a study in mice showed that myricetin, a dietary antioxidant, improved hepatic steatosis, and this was associated with increased hepatic type 1 deiodinase activity [124]. Moreover, the literature suggests that hepatic fatty acids in NAFLD may impair TH receptor activity [125]. The local hypothyroid status decreases hepatic lipase activity, which promotes triglyceride accumulation [126,127,128]. Furthermore, animal-based studies in models of MS and NAFLD have shown that the administration of both TH and TH agonists ameliorates hepatic steatosis [126,127,128]. Although further studies are needed, human data are beginning to appear. Low-dose TH therapy in humans leads to safe and effective results in NAFLD patients [129]. A recent randomized double-blind placebo-controlled trial with a selective thyroid hormone receptor-β agonist showed a significant decrease in hepatic fat content in patients with NASH [130]. 

Concerning systemic TH, Liu et al. showed that both free triiodothyronine (FT3) and TSH levels were positively correlated with the risk of NAFLD in euthyroid individuals [131]. Van der Bergh et al. also studied this topic on a euthyroid population, and the results evidenced that NAFLD patients presented with higher FT3 and lower free thyroxin (FT4) levels, and no differences were recorded concerning TSH [132]. In a recent meta-analysis, Guo et al. concluded that TSH levels may be positively correlated with NAFLD, independent of TH levels, and that TSH levels increase with the progression of NAFLD [117]. This association was also recently observed by our group in a morbidly obese population [133]. Although there are numerous other studies in agreement with this line of thought, some studies argue that no correlation exists [134].

Since both overt and subclinical hypothyroidism are widely associated with cardiovascular risk factors that are, in turn, associated with NAFLD [135], this association may be the result of confounding indirect effects. However, the Rotterdam study evidenced a close relationship between NAFLD and overt hypothyroidism, independent of other metabolic risk factors [136].

The data from Kowalik MA et al. is also very interesting; these authors explored the recent hypothesis of an association between hepatocellular carcinoma (HCC) and hypothyroidism and showed that the administration of T3 compromised HCC progression, even when given at late stages, lifting the veil for a possible approach to this poor-prognosis-associated carcinoma [137].

In contrast to the considerable evidence already available in hypothyroidism, the effect of hyperthyroidism in NAFLD has been considerably less well-studied. In a case report of a patient with NASH in whom Graves’ disease (GD) was installed, the liver enzyme levels improved after the onset of GD and consequent hyperthyroidism and worsened after starting the treatment and returning to an euthyroid state [138]. Therefore, it seems that TH may stand out in a way that, even in pathological states, they seem to exert positive effects regarding NAFLD. Even if this is in line with the previously hypothesized role of TH in NAFLD pathophysiology, more studies are needed to confirm.

## 5. Reproductive System Dysfunction and NAFLD

It is widely appreciated that disorders causing sex hormone dysfunction may predispose patients to the development of MS [139,140,141]. Therefore, it is plausible that changes in the reproductive axis may lead to the development of NAFLD (Figure 4). 

Sex hormones play a central role in regulating glucose metabolism and insulin sensitivity [142]. The decrease in insulin sensitivity, particularly in the liver, leads to an increase in hepatic gluconeogenesis, glucose production, and lipogenesis, which may exacerbate hepatic steatosis [143]. Several studies suggest that sex hormones are, in part, responsible for liver energy homeostasis and the regulation of hepatic fat accumulation [144]. In association with metabolic dysfunction, a sex hormone imbalance may be responsible for inducing hepatic injury. Estrogens and androgens are present in both men and women, although in different concentrations. A growing body of evidence suggests that estrogens have important metabolic functions in both women and men [145]. Estrogens seem to play a protective role against hepatic fat accumulation by promoting lipolysis and decreasing lipogenesis, mostly through the induction of acetyl-CoA carboxylase and, consequently, improving mitochondrial function [146]. Estrogens also maintain the hepatic cholesterol balance, suppressing its synthesis and promoting cholesterol clearance. Namely, estrogen receptor signaling is essential for appropriate hepatic lipid metabolism and the evidence supports that both estrogen receptor α and the G protein-coupled estrogen receptor are important regarding this matter [147]. Studies in animal models have shown that androgens also have an important role in protecting the liver from fat accumulation and in decreasing the risk of NAFLD development [146]. On the contrary, in women, higher androgen levels can increase the risk of NAFLD risk, as described later in the polycystic ovary syndrome (POCS) subhead [148]. These opposite described effects might be explained by sex differences in androgenic action. In fact, in men, normal serum androgen levels prevent hepatic lipid accumulation, whereas androgen deficiency is associated with fatty liver accumulation [149]. A possible explanation is that serum testosterone concentration in these two spectrums (females with excess androgen and males with androgen deficiency) may overlap. This range of androgens might be the concentration window responsible for metabolic dysfunction, and some authors even designate this adverse concentration range of circulating androgen as the “metabolic valley of death” [150]. However, the cellular and systemic mechanisms of this phenomenon are still poorly understood.

### 5.1. Hypogonadism

Hypogonadism is defined as the reduction of sex hormones. There are various pathophysiological mechanisms underlying hypogonadism, but the presence of NAFLD seems to be an ubiquitous comorbidity in these patients [151]. The association between hypogonadism and NAFLD seems to be bidirectional, and causality is difficult to establish. There is a positive association between lower levels of testosterone and the prevalence of NAFLD [152,153]. Men with biopsy-proven NASH have diminished dehydroepiandrosterone (DHEA) levels, and this decrease parallels an increasing severity of hepatic fibrosis [115,153,154] DHEA modulates oxidative stress sensitivity, as it is a potential mediator of reactive oxygen species scavengers, enhances insulin sensitivity, and increases peroxisome proliferator-activated receptor α expression [155]. Serum DHEA levels might be used for the identification of patients at risk for the development of NASH with advanced fibrosis [154]. 

Additionally, the literature advocates that hypogonadism is associated with important cardiovascular risk factors, such as general and visceral obesity, impaired insulin sensibility, hypertension, and dyslipidemia [115]. These are also crucial contributors to the development of NAFLD and can be potential confounders in some studies in this topic. A study by Sakr et al. evidenced that male rats with decreased gonadal hormones have higher serum and hepatic levels of triglycerides, cholesterol, and LDL and have increased levels of enzymes involved in hepatic steatogenesis, such as sterol regulatory element-binding transcription factor 1 (SREBP-1) and 2 (SREBP-2) [156]. Androgen receptor knockout male mice acquire late-onset obesity and have an increased risk of NAFLD development and progression [157]. Hormonal replacement therapy (HRT) with testosterone seems to ameliorate these changes and may be considered a protective strategy to be taken into account [158]. Clinical studies show that long-term testosterone replacement improves MS components and ameliorates liver enzymes changes in men with hypogonadism [158]. 

Regarding women, the most common cause of primary hypogonadism is Turner syndrome. This syndrome is characterized by a premature ovarian failure that causes estrogen deficiency [159]. Hypoestrogenism has an important role in some of the most common comorbidities of these patients, such as bone mineralization disorders and hepatic dysfunction, with increased hepatic enzymes [160,161]. Besides other benefits, HRT may improve hepatic function and other components of metabolic dysfunction in women with Turner’s syndrome. Animal studies are also in line with this association. A study showed that estrogen deficiency worsens the hepatic histological injury in mice fed a high-fat and high-cholesterol diet [162]. In mice fed a high-fat and high-cholesterol diet, lower estrogen levels were associated with higher levels of macrophage infiltration and the enhanced expression of hepatic inflammatory genes, and HRT was able to reverse these changes [162]. Both human and animal studies evidence the benefits of HRT on hepatic function and other components of metabolic dysfunction [160,163]

### 5.2. Impact of Menopause on Liver Disease

As discussed throughout this manuscript, an imbalance in estrogen levels might have a pronounced impact on hepatic homeostasis. Menopause is a physiological condition of estrogen deficiency with an important impact on women’s health. The risk of development and progression of NAFLD increases with the duration of estrogen deficiency [164]. Accordingly, women with premature menopause are at increased risk of severe liver fibrosis [164]. Thus, both menopausal status and menopause onset age should be taken into account when determining fibrosis risk among women with NAFLD. An epidemiological study evidenced a higher prevalence of NAFLD in postmenopausal female patients when compared with men [165]. Another study found a prevalence of NAFLD of approximately 60% in postmenopausal women, compared to 32% in premenopausal patients [166]. The increased risk of NAFLD is mainly caused by estrogen deficiency, similar to women with oophorectomy or premature ovarian failure [167]. Moreover, women with NAFLD have a higher risk of fibrosis progression as they age [168]. Postmenopausal women have a higher severity of fibrosis at any given hepatocyte ballooning and portal inflammation [169]. The fat redistribution associated with menopause increases the risk of insulin resistance, dyslipidemia, hypertension, and diabetes and, consequently, may increase the risk of NAFLD [170]. Among post-menopausal women, HRT is associated with a reduced risk of NAFLD and fibrosis progression. The administration of HRT among post-menopausal women appears to be protective against NAFLD development, but whether it affects fibrosis progression is still unclear [164]. Further studies evaluating the potential impact of HRT on hepatic fibrosis in postmenopausal women are needed, including the adequate dose, route of administration, duration, and age of initiation.

### 5.3. Polycystic Ovary Syndrome (PCOS)

Concerning diseases with increased levels of sex hormones, it is important to address PCOS. Recent guidelines define PCOS as a clinical and/or biochemical hyperandrogenism, chronic oligo-anovulation, and polycystic ovarian morphology [171]. The diagnosis requires two out of three of these criteria after the exclusion of other endocrine disorders [171]. A high percentage of women with PCOS present with obesity and MS [172]. Evidence suggests that the prevalence of NAFLD is increased in women with PCOS, regardless of weight and MS [173]. The prevalence of NAFLD in women with PCOS is 35 to 70%, compared with 20 to 30% in age- and body mass index (BMI)-matched control women [173]. PCOS is a prevalent condition among patients with biopsy-confirmed NAFLD (approximately 50–70%), and the risk of NAFLD development in POCS patients is two-fold higher compared with control women [174]. Women with POCS are also more likely to have more severe histological features, such as NASH, advanced fibrosis, and cirrhosis [175]. The pathophysiological mechanism underlying the increased risk of NAFLD in PCOS is thought to be multi-factorial. Both genetic and acquired factors are involved, with contributions from abdominal adiposity, systemic insulin resistance, chronic inflammation, and hyperandrogenism [173]. Whether NAFLD is associated with POCS as a consequence of shared risk factors or PCOS independently supports NAFLD development remains to be elucidated. Hyperandrogenism may be considered a central contributor to NAFLD development [174,176,177] Jones et al. evidenced that women with POCS and higher androgen levels have greater intra-hepatic fat content compared with women with POCS and lower androgen levels [178]. A recent meta-analysis showed that women with POCS but without hyperandrogenism were not associated with an increased prevalence of NAFLD [174]. Increased levels of testosterone in women can enhance lipogenic gene expression and de novo lipogenesis in hepatocytes. Therefore, the liver injuries seen in NAFLD may also be caused by lipotoxicity, which is commonly present in women with POCS [149]. Qu et al. stated that women with PCOS and NAFLD had higher BMI, waist/hip ratio, worse insulin resistance, increased triglycerides, and lower HDL cholesterol levels than women with POCS without NAFLD. Moreover, co-existing NAFLD seems to worsen the metabolic profile of women with POCS [179].

Current guidelines for the treatment of POCS consist of lifestyle intervention. When a pharmacologic approach is needed to improve the metabolic profile, metformin remains the drug of choice [180]. However, metformin does not seem to affect liver histology in patients with NAFLD and PCOS [181]. Oral contraceptive pills (OCP) increase serum lipid levels and, consequently, might worsen NAFLD [182]. Interestingly, according to Liu SH. et al., women using OCP have a lower incidence of NAFLD, but this might be mediated or confounded by adiposity [183]. There are no studies evaluating the impact of OCP on the liver profile in women with PCOS. Preliminary results show that liraglutide and other glucagon-like peptide-1 receptor agonists can decrease the intra-hepatic fat content and visceral adipose tissue among obese women with PCOS [181]. Additionally, the prevalence of NAFLD was reduced by two thirds in obese women with PCOS treated with liraglutide [181]. More accurate diagnostic methods, such as liver biopsy, in NASH patients have evidenced a reduction in hepatic inflammation and the prevalence of NASH during liraglutide treatment [184]. However, there is still not enough evidence to support the use of these drugs in these patients. More studies are needed to evaluate whether treatment with anti-androgenic drugs may reduce the risk of NAFLD in women with PCOS. 

### 5.4. Important Sex Differences in NAFLD Development and Severity

As mentioned earlier, sex hormones have an important impact on the pathophysiology and progression of NAFLD and should be taken into account when studying this topic [185,186]. Sex differences and other factors, such as dietary patterns, exercise, and quality of life, are equally important to consider when studying NAFLD [187]. The understanding of the impact of sex in NAFLD remains scarce. Only a few studies describe sex differences in NAFLD, even though some of the major risk factors for NAFLD are profoundly impacted by sex. This hepatic disorder is more prevalent and has worse histological features in men at reproductive ages. The difference fades after menopause, when the prevalence of NAFLD in women equals or exceeds that of men, as discussed earlier [169]. Moreover, sex differences have a significant impact on hepatic outcomes in patients with biopsy-confirmed NAFLD and advanced hepatic fibrosis, with a higher incidence of HCC and a worse survival rate in male patients compared with premenopausal women [188,189]. Men with NASH cirrhosis have an almost 2-fold to 7-fold higher risk of having HCC than women [190]. It seems that men develop HCC at earlier liver fibrosis stages than women, which may explain the higher prevalence of HCC in men [191]. Finally, animal studies evidenced that male mice have innate immune cells that promote liver inflammation and fibrosis, while macrophages from female mice have a more protective and antifibrotic phenotype [192]. In fact, Kupffer cells from male animals produce more IL-6, which increases inflammation, tissue injury, and subsequent hepatocyte proliferation when compared with female animals. Furthermore, estrogen has an inhibitory effect on IL-6 production, stellate cell activation, and fibrogenesis [193]. Stellate cells incubated with estrogen have a significant decrease in collagen synthesis, which partially explains the delay in the progression of liver fibrosis in pre-menopausal or post-menopausal women treated with HRT [194]. Evidence proves that males and females respond differently to lifestyle modifications and drug therapies [195]. Therefore, a patient’s sex and hormonal status are crucial factors to take into account when studying NAFLD pathophysiology and management. 

Regardless of menopausal status, serum adiponectin is higher in women than men [196,197]. Serum adiponectin levels are inversely associated with body weight and central obesity and have been proposed as a biomarker of MS [198]. Data regarding the beneficial effects of adiponectin on NAFLD are well documented. Adiponectin can reduce insulin resistance and has anti-steatotic and anti-inflammatory effects [199,200]. There is an association between lower levels of adiponectin and an increased risk and severity of NAFLD [199,200]. Although the causality of this association is difficult to establish, adiponectin may be considered a biomarker of NAFLD progression to NASH and cirrhosis [201]. A recent systematic review showed that the administration of pioglitazone increases serum adiponectin levels and improves histological features in NASH patients [202]. Although pioglitazone has been associated with an increased risk of heart failure, the use of selective PPARγ modulators may be promising [203]. A deeper knowledge of the dynamic crosstalk between adipokines and the liver may result in a better and personalized treatment option in the future.

## 6. Adrenal Gland Disorders and NAFLD

### 6.1. Renin–Angiotensin–Aldosterone System (RAAS)

While there are some studies in animal models and/or patients with primary aldosteronism, most research has been conducted in the context of hyperstimulation of the RAAS. Kumagai et al. found that hyperaldosteronism results in the development of insulin resistance in patients with previously normal insulin metabolism (10-year follow-up) [204]. Activation of the RAAS leads to altered insulin/IGF-1 signaling pathways in several tissues, namely, the liver [205]. Local hepatic increased insulin resistance may lead to inadequate lipid accumulation and, eventually, to NAFLD [206]. The association between an inappropriately active RAAS and insulin resistance in the setting of the MS is an area of growing interest [207]. Although aldosterone has been more extensively studied in the context of NAFLD, growing evidence suggests other hormones closely related to aldosterone may have significant and possibly independent roles in the development of NAFLD. Both angiotensin II and the angiotensin II type 1 receptor and mineralocorticoids and mineralocorticoids receptor activation contribute to insulin signaling attenuation [205,207]. Moreover, RAAS-mediated reactive oxygen species formation not only appears to play a vital role in insulin signaling impairment but also leads to endothelial dysfunction, which contributes to hypertension, atherosclerosis, chronic kidney disease, and cardiovascular disease [205,207]. Additionally, even ionic disturbances, such as the hypokalemia that is found in overactive RAAS states, seem to contribute to NAFLD [208]. Whether they do so in a synergistic or independent manner remains unknown.

Noguchi et al. showed that selective aldosterone blocker (SAB) treatment inhibited liver fibrogenesis and carcinogenesis by suppressing neovascularization and activated hepatic stellate cells in a NASH rat model [209]. Polyzos et al. showed in a preliminary report that spironolactone improved insulin resistance in patients with NAFLD [210]. Another similar study by Pizarro et al. found that eplerenone effectively ameliorated histological steatosis and hepatic fibrosis in a mouse model of NASH, corroborating the aforementioned evidence [211]. Hence, aldosterone plays an important role in the progression of NASH and SAB, which are widely used and safely represent a potential new therapeutic strategy for NAFLD/NASH. Spironolactone may have anti-inflammatory and anti-fibrogenic effects on the liver and, therefore, may be an inexpensive and effective therapeutic target for NAFLD. However, a large-scale human trial is needed to further explore this hypothesis. 

Most of the evidence sustaining that aldosterone has an important role in the development of NAFLD is based on its effects on insulin resistance. One study examined aldosterone’s direct and independent effects in NAFLD. Kumar et al. found a positive association between serum aldosterone levels and fatty liver in African American women [212]. The association was independent of alcohol intake and persisted after insulin resistance and high-sensitivity C-reactive protein were added to the multivariable analysis [212]. These results suggest that the effect of aldosterone in fatty liver is partially independent of insulin resistance and high-sensitivity C-reactive protein. Although the results are promising, a few limitations of the study should be considered, namely, the inability to prove causality and the need to study different ethnicities and to clarify sex differences with larger samples. Similarly, a study by Lu et al. was able to isolate the angiotensinogen effects from those of angiotensin II and renin and found it contributes to body weight gain and liver steatosis [213].

Angiotensin II receptor blockers are generally considered to have no significant metabolic effects. However, experimental and clinical evidence suggest telmisartan is an exception. Besides blocking the angiotensin II type 1 receptor, telmisartan is a selective PPARγ modulator [214]. Telmisartan’s selectiveness results in the well-established therapeutic benefits of PPARγ modulation in lipid and glucose metabolism without the undesirable side effects of conventional PPARγ activators, such as fluid retention, weight gain, and edema [214]. Moreover, the routine hypertension treatment dosage is enough to obtain the aforementioned effects [214]. Hence, telmisartan could play an important role in the prevention and treatment of MS, diabetes, and atherosclerosis [214]. A randomized controlled study by Hirata et al. explored the effect of telmisartan and losartan in improving steatosis in hypertensive NAFLD patients with type 2 diabetes [215]. While significant liver function improvement was not found in either group, the serum free fatty acid levels were significantly decreased in the telmisartan group. In addition, telmisartan also improved the liver/spleen ratio. By improving hepatic fat deposition, telmisartan may be a potential new therapeutic strategy for NAFLD [215]. Yokohama et al. showed that treatment with losartan improved not only serum liver enzyme levels but also reduced hepatic necroinflammation and fibrosis in patients with NASH [216]. More specifically, losartan decreased transforming growth factor beta 1(TGF-β1) plasma levels, a known marker of hepatic fibrosis. The authors of the study hypothesized that angiotensin II receptor antagonists may be a safe and efficacious therapeutic option for NASH [216]. 

Fallo et al. showed that patients with primary hyperaldosteronism and hypokalemia were at a greater risk of developing metabolic and liver disease [217]. A higher prevalence of NAFLD and insulin resistance was found in the hypokalemic patient subgroup compared to those with normokalemia [217]. Sun et al. found that hypokalemia was independently associated with the prevalence of MS in a large Chinese population and hypothesized that potassium-correcting therapies could alter the progression of this disease [218]. 

### 6.2. Cushing’s Syndrome and NAFLD

Glucocorticoids (GC) are produced by the adrenal gland under the control of ACTH secreted by pituitary. The effects of GC on lipid metabolism, fat accumulation, and NAFLD development are complex (Figure 5). Furthermore, hepatic dysfunction may impair GC metabolism and alter the adrenal axis. A small study with 50 patients reported NAFLD in 20% of patients with Cushing’s syndrome, results similar to a retrospective study with a prevalence between 26 and 33% [219,220]. Additionally, Rockall et al. found that 20% of patients with Cushing’s syndrome had hepatic steatosis in a computed tomography scan [219]. Contrariwise, patients with NAFLD did not present with Cushing’s syndrome, though some abnormalities of GC metabolism, such as increased urinary free cortisol and decreased dexamethasone suppression of plasma cortisol, have been shown [221,222]. GC may contribute to the development of NAFLD through their lipolytic effects on adipose tissue, which result in more readily available free fatty acid for liver uptake [8]. The association between GC and hepatic lipid accumulation has been observed in both clinical and basic research. Auer et al. found a daily hydrocortisone dose to be positively associated with hepatic lipid accumulation in humans [223], and D’Souza A et al. showed that exogenous corticosterone treatment in rats induced hepatic steatosis [224]. Targher et al. found chronic hypothalamic–pituitary–adrenal (HPA) axis hyperactivity and subclinical hypercortisolemia in NAFLD patients [222]. In a more recent study, Hayashi et al. hypothesized that activation of adipocyte GC receptors by high levels of GC in patients with Cushing’s syndrome generates a cascade of biomolecular events, culminating in hepatic steatosis and insulin resistance [225]. By accelerating lipolysis and restricting healthy adipose expansion, adipocyte GC receptor activation leads to ectopic lipid accumulation and insulin resistance. This study suggests that the adipocyte GC receptor is a promising target for the treatment of metabolic dysfunction/diseases present in Cushing’s syndrome [225]. It is important to acknowledge that cortisol’s influence on adipose tissue is dependent on insulin status. During fasting, insulin levels are low, and cortisol stimulates lipolysis. Paradoxically, when insulin levels are high, cortisol stimulates lipogenesis [226]. Furthermore, it has been hypothesized that sympathetic nervous system activation may underlie the hypercortisolemia commonly observed in NAFLD patients through the stimulation of cortisol release [226]. 

One enzyme of the GC cascade has sparked great interest in this field. 11β-hydroxysteroid dehydrogenase type 1 (11β-HSD1) catalyzes the conversion of cortisone into cortisol. Chronic hypercortisolemia, obesity, and NAFLD increase local adipose tissue levels of cortisol by upregulating the expression of 11β-HSD [227]. Studies have shown a decrease in hepatic steatosis when 11β-HSD1 is pharmacologically inhibited or genetically absent [228,229]. The role of A-ring reductases (5aR and 5bR), which inactivate cortisol, has not yet been thoroughly studied [8]. However, a study found an increased risk of hepatic steatosis with the deletion of 5aR type 1 [230].

Interestingly, one study by Ahmed et al. has defined two seemingly protective phases of altered hepatic cortisol metabolism in progressive NAFLD [231]. In steatosis, increased cortisol clearance leads to lower local levels of this hormone, consequently preserving the hepatic metabolic phenotype and limiting lipid accumulation. On the other hand, increased cortisol regeneration and, therefore, higher local cortisol levels are present in NASH, possibly to limit hepatic inflammation. This distinction is particularly pertinent when looking at 11β-HSD1 as a potential therapeutic target. Inhibition of 11β-HSD1 might be favorable in steatosis since it would further reduce local levels of cortisol. However, 11β-HSD1 inhibition in NASH could be detrimental, as it would worsen the inflammatory response. Therefore, the histological stage of NAFLD may dictate whether 11β-HSD1 inhibition is beneficial.

### 6.3. Pheochromocytoma

Human primary hepatic stellate cells are dependent on catecholamines for their survival and fibrogenic effects [232]. Sigala et al. studied these cells and reported an upregulation of fibrogenic α/β-adrenoreceptor and neuropeptide Y receptors in human cirrhotic NAFLD [232]. Furthermore, the authors suggested adrenoreceptor and neuropeptide Y antagonists as potential anti-fibrotic agents for NAFLD. Although the effects of excessive systemic catecholamines levels are still widely unknown, one can speculate that patients with pheochromocytoma may be at higher risk of NAFLD based on the results of this study. More studies are necessary to further clarify the role of catecholamines and their related disorders in NAFLD.

## 7. Conclusions

The prevalence of NAFLD is rising and the understanding of its pathophysiology is crucial for better management of this condition. It is biologically plausible that there is a link between several endocrine disorders and NAFLD, other than the typically known type 2 diabetes mellitus, obesity, and MS. Namely, our review presents that there is evidence supporting that hyperprolactinemia, excessive or defective GH, hypothyroidism, hypercortisolism, activation of the RAAS, PCOS, and hypogonadism are endocrine disorders that exert a negative/facilitative effect on NAFLD initiation or progression. The contribution of the other endocrine entities that we explored remain fairly controversial or uncertain. Table 1 and Figure 6 summarize the main conclusions of the studies included in this review. 

It is important to keep in mind when seeing patients with endocrine diseases that NAFLD may develop or that a pre-existing condition may exacerbate it with immeasurable consequences. Furthermore, NAFLD is a heterogeneous metabolic disorder, and we agree that the recognition of such associations is crucial towards the implementation of a tailored approach to our patients [233,234]. As described throughout the text, a better understanding of the relationship between these hormonal imbalances and the development and progression of NAFLD can lead to treatment advances. Nevertheless, controversial and insufficient evidence in this area of knowledge preclude us from drawing definite conclusions. Prospective and well-designed studies are lacking but are crucial and must be carried out in the near future for a better understanding and treatment of patients with NAFLD.

## Figures and Tables

**Figure 1 metabolites-12-00298-f001:**
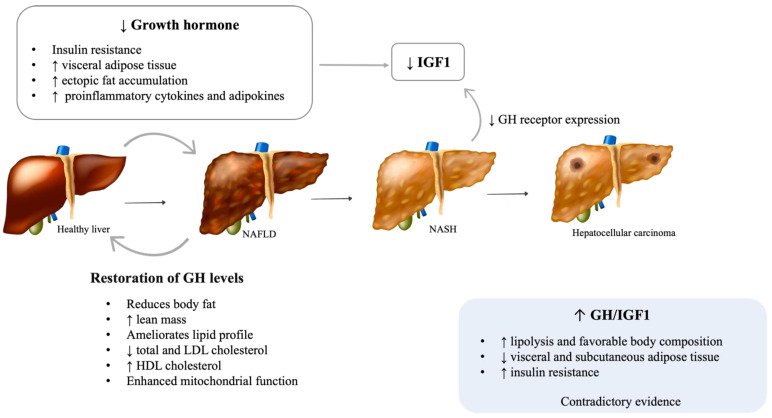
Possible mechanisms of GH deficiency on NALFD development and progression. NAFLD: Nonalcoholic fatty liver disease; NASH: Nonalcoholic steatohepatitis; GH: Growth hormone; IGF1: Insulin-like growth factor 1. [Viktoriia Kasyanyuk] © https://www.123rf.com/ (accessed on 22 February 2022) and Servier Medical Art by Servier.

**Figure 2 metabolites-12-00298-f002:**
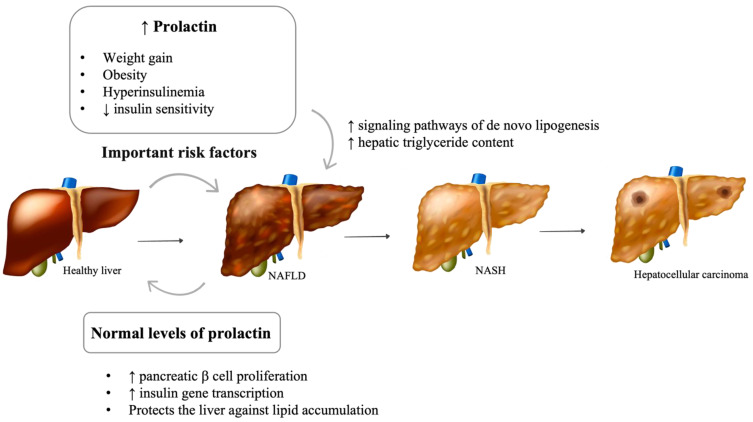
Impact of prolactin levels on NAFLD development and progression. NAFLD: Nonalcoholic fatty liver disease; NASH: Nonalcoholic steatohepatitis; GH: Growth hormone; IGF1: Insulin-like growth factor 1. [Viktoriia Kasyanyuk] © https://www.123rf.com/ (accessed on 22 February 2022) and Servier Medical Art by Servier.

**Figure 3 metabolites-12-00298-f003:**
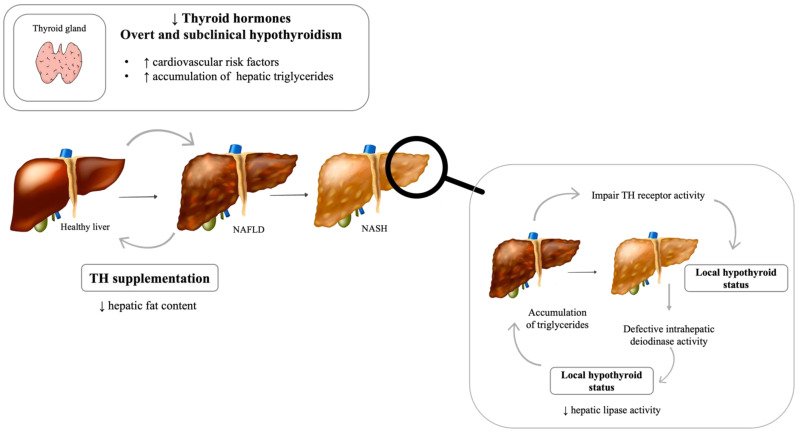
Thyroid–liver axis: The impact of thyroid hormones on liver. NAFLD: Nonalcoholic fatty liver disease; NASH: Nonalcoholic steatohepatitis; TH: Thyroid hormone. [Viktoriia Kasyanyuk] © https://www.123rf.com/ (accessed on 22 February 2022) and Servier Medical Art by Servier.

**Figure 4 metabolites-12-00298-f004:**
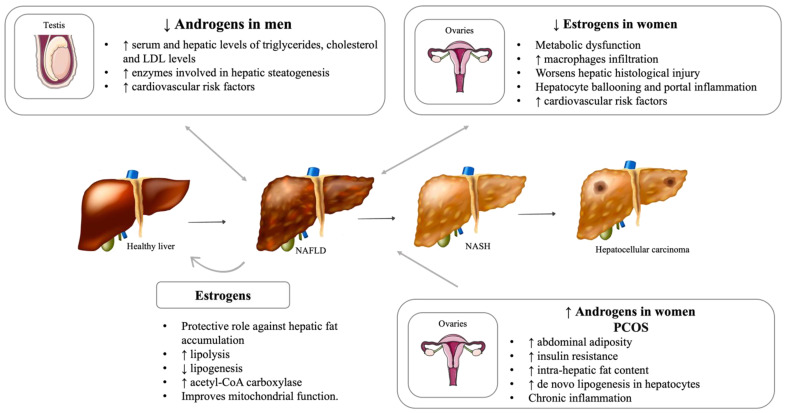
Sexual hormones and NAFLD. NAFLD: Nonalcoholic fatty liver disease; NASH: Nonalcoholic steatohepatitis; PCOS: Polycystic ovary syndrome. [Viktoriia Kasyanyuk] © https://www.123rf.com/ (accessed on 22 February 2022) and Servier Medical Art by Servier.

**Figure 5 metabolites-12-00298-f005:**
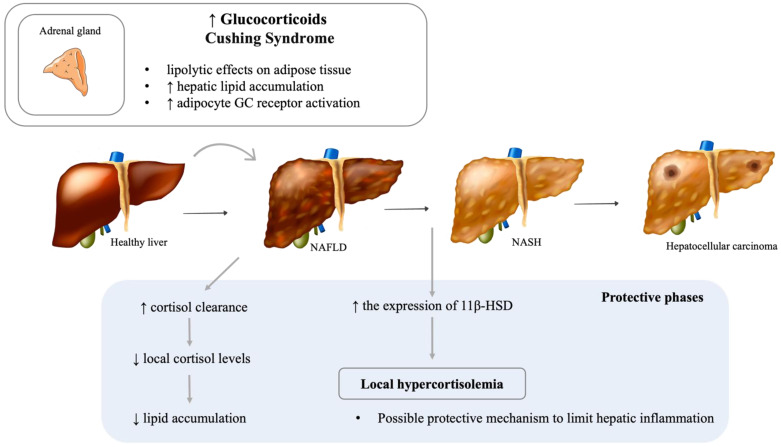
Impact of cortisol imbalance and NAFLD pathogenesis. NAFLD: Nonalcoholic fatty liver disease; NASH: Nonalcoholic steatohepatitis. [Viktoriia Kasyanyuk] © https://www.123rf.com/ (accessed on 22 February 2022) and Servier Medical Art by Servier.

**Figure 6 metabolites-12-00298-f006:**
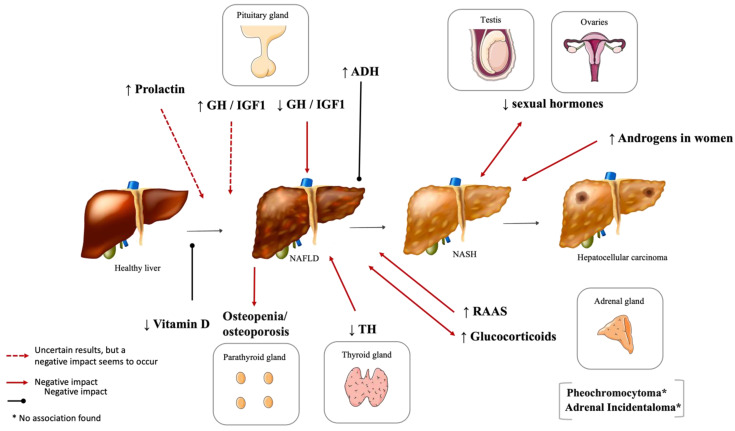
Summary figure of the crosstalk between the endocrine axes and NAFLD. NAFLD: Nonalcoholic fatty liver disease; NASH: Nonalcoholic steatohepatitis; GH: Growth hormone; IGF1: Insulin-like growth factor 1; ADH: antidiuretic hormone; PCOS: Polycystic ovary syndrome; TH: Thyroid hormones; RAAS: Renin–Angiotensin–Aldosterone System. [Viktoriia Kasyanyuk] © https://www.123rf.com/ (accessed on 22 February 2022) and Servier Medical Art by Servier.

**Table 1 metabolites-12-00298-t001:** Effect of endocrine hormones on the different factors involved in nonalcoholic fat liver disease development.

Hormone	Plasmatic Levels	Effect	Human Studies	Animal Studies	In Vitro Studies
Growth hormone/IGF1	Increased	↑ lipolysis and lean body mass	Maison, Griffin et al. 2004, Petrossians, Daly et al. 2017		
		↓ visceral and subcutaneous adipose tissue	Petrossians, Daly et al. 2017, Koutsou-Tassopoulou, Papapostoli-Sklavounou et al. 2019		
		Impairment of lipid profile and ↑ insulin resistance	Moller and Jorgensen 2009		
	Decreased	Hepatic steatosis	Ichikawa, Hamasaki et al. 2003, Laron, Ginsberg et al. 2008, Nishizawa, Iguchi et al. 2012, Nguyen, Ricolfi et al. 2018	Sos, Harris et al. 2011, Nishizawa, Takahashi et al. 2012, Cordoba-Chacon, Majumdar et al. 2015	
		↓ Hepatic inflammation and fibrosis	Sumida, Yonei et al. 2015		
		↑ proinflammatory cytokines and adipokines	Ukropec, Penesova et al. 2008		
	Restoration of GH levels	↑ lean body mass and ↓ body fat	Newman, Carmichael et al. 2015		
		Improvement of lipid profile	Newman, Carmichael et al. 2015		
		Improvement of hepatic injury	Nishizawa, Iguchi et al. 2012, Takahashi, Iida et al. 2007	Nishizawa, Takahashi et al. 2012	
Prolactin	Increased	Weight gain	Serri, Li et al. 2006		
		↓ insulin sensitivity and hyperinsulinemia	Serri, Li et al. 2006		
		↑ hepatic triglyceride content and de novo lipogenesis		Park, Kim et al. 2011, Luque, Lopez-Vicchi et al. 2016	
	Decreased	↓ hepatic cholesterol and triacylglycerol	Zhang, Ge et al. 2018		
		↓ expression of genes involved in lipogenesis Hepatic steatosis	Zhang, Ge et al. 2018		
	Restoration of prolactin levels	Weight loss	Berinder, Nystrom et al. 2011		
		↑ insulin sensitivity and glycemic control	dos Santos Silva, Barbosa et al. 2011		
		Improvement of lipid profile	dos Santos Silva, Barbosa et al. 2011		
Vitamin D	Normal/Increased	↓ hepatic fibrogenesis			Beilfuss, Sowa et al. 2015
		Anti-inflammatory, anti-proliferative, and anti-fibrotic effects in liver			Abramovitch, Dahan-Bachar et al. 2011
Thyroid Hormones	TH supplementation	↓ Hepatic steatosis	Bruinstroop, Dalan et al. 2018, Harrison, Bashir et al. 2019	Erion, Cable et al. 2007 Cable, Finn et al. 2009, Mollica, Lionetti et al. 2009	Erion, Cable et al. 2007
	Decreased	↓ hepatic lipase activity and ↑ triglyceride accumulation		Erion, Cable et al. 2007 Cable, Finn et al. 2009, Mollica, Lionetti et al. 2009	Erion, Cable et al. 2007
Estrogens	Increased	Improves hepatic function		Chambliss, Barrera et al. 2016: ♀	
		↓ IL-6 levels		Naugler, Sakurai et al. 2007: ♂	
		↓ collagen synthesis in stellate cells			Itagaki, Shimizu et al. 2005
	Decreased	Worsens the hepatic histological injury, ↑ macrophage infiltration, and enhanced expression of hepatic inflammatory genes		Kamada, Kiso et al. 2011	
	Estrogen replacement therapy	Ameliorates hepatic histological injury		Kamada, Kiso et al. 2011	
		↓ macrophage infiltration and enhanced expression of inflammatory genes		Kamada, Kiso et al. 2011	
		Improves hepatic function	Elsheikh, Hodgson et al. 2001♀		
Androgens	Decreased levels (♂)	↑ serum and hepatic levels of triglycerides, cholesterol, and LDL		Sakr, Hussein et al. 2018	
		↑ enzymes involved in hepatic steatogenesis (SREBP-1 and SREBP-2)		Sakr, Hussein et al. 2018	
		↑ hepatic fat accumulation and NAFLD development and progression		Fan, Yanase et al. 2005	
		↑ late-onset obesity		Fan, Yanase et al. 2005	
	Increased levels (♀)	↑ intra-hepatic fat content	Wang, Guo et al. 2020		
	Testosterone replacement therapy	Improvement of metabolic syndrome	Traish, Haider et al. 2014		
RAAS	Increased	↑ insulin resistance	Kumagai, Adachi et al. 2011)		
		↑ ROS formation and endothelial dysfunction	Chen, Muntner et al. 2003		
		↑ weigh gain and hepatic steatosis		Lu, Wu et al. 2016	
	Decreased/Pharmacological block	↓ hepatic fibrogenesis and carcinogenesis		Noguchi, Yoshiji et al. 2010	
		↓ insulin resistance in NAFLD	Polyzos, Kountouras et al. 2011		
		↓ Steatosis and hepatic fibrosis	Hirata, Tomita et al. 2013	Pizarro, Solis et al. 2015	
		↓ serum free fatty acid level and improved the liver/spleen ratio	Hirata, Tomita et al. 2013		
		↓ serum liver enzyme and TGF-1 levels, ↓ hepatic necroinflammation	Yokohama, Yoneda et al. 2004		
Glucocorticoids	Increased	Hepatic steatosis		D’Souza A, Beaudry et al. 2012, Auer, Stalla et al. 2016	
		Insulin resistance			Hayashi, Okuno et al. 2019
		Possibly limits hepatic inflammation in NASH patients	Ahmed, Rabbitt et al. 2012		
Catecholamines	Normal/Increased	↑ Fibrogenic α/β-adrenoreceptor and neuropeptide Y receptors in human cirrhotic NAFLD			Sigala, McKee et al. 2013

Legend: NAFLD: Nonalcoholic fatty liver disease; NASH: Nonalcoholic steatohepatitis; IGF1: Insulin Like Growth Factor 1; RAAS: renin–angiotensin–aldosterone system; TGF-1: transforming growth factor beta 1; ROS: reactive oxygen species; IL-6: Interleukin 6. Text in italic represents controversial results. gender: ♀ female, ♂ male.

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
