# Peer review of "Nonalcoholic Fatty Liver Disease and Endocrine Axes—A Scoping Review"

_metabolites, 2022, doi:10.3390/metabo12040298_

Round 1
Reviewer 1 Report
Here the authors describe the possible association between the non-alcoholic fatty liver disease (NAFLD) and a wide spectrum of endocrinopathies. They explore the crosstalk between the liver and different kinds of endocrine organ dysfunctions, excluding the diabetes mellitus, obesity and metabolic syndrome, which are deeply explored in other works.
In my opinion, the paper is generally well written and proposes new insights in the field of endocrinopathies that induces NAFLD when compared with paper already published. I appreciated the insight about acromegaly, hyperprolactinemia, vasopressin, disturbances of bone metabolism and pheochromocytoma. However, there are some points to note that I listed below.
- I think it is very important to mention the review by Marino et al. 2015 (PMID 26494962) and Singeap et al 2021 (PMID: 33505943) as they explore the same issues dealt with here by the authors.
- The author with affiliation number 4 is missing from the list of authors.
- Many oversights, spelling and punctuation errors are scattered throughout the text and the table 1. I suggest a minor revision of the English language and the correction of the oversights in the text and in the table 1.
- When authors introduce a new acronym please insert the full wording first and then the initials in brackets.
- There is a problem with the number of pages, please correct it.
- To better clarify some specific molecular mechanism, it could be useful to introduce some figures: for example, a schematic representation of the role of GH in de novo lipogenesis in the GH deficiency, or the authors can choose other significant mechanisms to represent.
Reviewer 2 Report
Thnak you so much for your valuable studies for contributing the knowledges between NAFLD (NASH) and endocrine system.
As a reviewer, I highly felt this kind of study should have to release and emphasize the above fileds.
However, to publish this paper there are serious issues on the current manuscript. Please see the below and re-try to submission upcoming futures.
This time this reviewer's decision is Revision.
Major concerns:
Abstract :
Authors tried to re-describe abstract section. The current version is very simple and generized. Please, be clearly mention about the disease and endocrine system and their relationship to develop NASH , briefly.
Introduction:
'Introduction' part is too short. This is a review paper which would be need to write with very professional opinion. But, your introducion is too short to fully understand the interaction between NAFLD and Endorcine-axis.
Please re-prepare the Table 1. This version is very vague and difficult to see it.
Through the manuscript:
Please, insert brief vesion of figures according to the each of section. That would be much more helpful to understand your study.
Conclusion:
Please re-describe the concluison part accroding to the each of discovers with numbering.
Reviewer 3 Report
Manuscript ID: metabolites-1630595
Title: "Non-Alcoholic Fatty Liver Disease and Endocrine Axes – A 2 Scoping Review"
Authors: Madalena Von-Hafe et al.
The authors of the present review try to present the associations of the endocrine mechanisms, the related hormones, and disorders with the NAFLD. It is an interesting and relatively thorough review; however, the following points should be considered.
Comments:
- The authors should add a statement regarding the methodology, even to state that a systematic review according to PRISMA guidelines was not performed, or the study was not PROSPERO registered, and formal PICO search strategies are not available. However, if some type of literature search was performed, it should be described.
- The authors should also discuss the impact on oxidative stress and mitochondrial function.
- The authors should add a brief comment regarding the GPER (G-protein coupled estrogen receptor)
- The authors should add a figure legend to briefly describe the illustrated mechanisms. Also, the role of sex hormones should be demonstrated. Moreover, the figure should be reorganized in order to highlight the effects of the hormones and endocrine axes on each NAFLD stage and, secondly, the disease/disorders related to the hormonal disturbances.
- In the introduction, the authors mention that they will not include the associations between NAFLD with T2D, metabolic syndrome, and obesity. However, they should clarify that they did not discuss the adipose and pancreas tissues as well as their related hormones and endocrine axes, which is a limitation of this review.
Reviewer 4 Report
GENERAL COMMENT
The gamut of interactions between the endocrine system and the liver spans across the spectrum from physiological to disease states. In particular, NAFLD/MAFLD is the prototypic condition of liver involvement in several endocrinopathies in humans. On these grounds, submission Metabolites – 1630595 identifies itself as a timely contribution which exhibits a large number of relevant bibliographic references. Therefore, my suggestions are aimed at improving this submission further.
SPECIFIC COMMENT
MAJOR
1. Introduction - should be reworked to a substantial extent to
a) expand on rationale and aims of this study. In doing so, it would be important to report on "secondary NAFLD forms" (J Hepatol. 2021 Jun;74(6):1455-1471). In this regard it is critical to define how far one should go in ruling out competing NAFLD causes (https://doi.org/10.37349/emed.2020.00007)
b) Additionally, I found Table 1 not to be fully appropriate for this section of the manuscript and should best be moved to elsewhere (probably in the concluding remarks).
c) Finally, I think a short section on methods should be included: describe which key words were used, which datatabase were consulted; and what the time frame of this literature research.
2. Growth Hormone
a) I would suggest expanding cited studied by adding Am J Gastroenterol. 2002 Apr;97(4):1071-2; Hepatology. 2014 May;59(5):1668-70. Liver Int. 2012 Feb;32(2):279-86.
b) Please, address case studies of GH deficiency in children submitted to hormone replacement therapy.
3. Sex Hormones
Discuss the role of androgens in additional detail (Hepatology. 2008 Feb;47(2):484-92).
4. Thyroid
I suggest some additional studies to discuss: Thyroid. 2018 Oct;28(10):1270-1284.Dig Liver Dis. 2019 Apr;51(4):462-470. J Hepatol. 2020 Jun;72(6):1159-1169.
5. Hepatocellular carcinoma (HCC)
It would be important to develop this topic further. The so called NAFLD-HCC poses a threaten to these patients. Further exhibiting a significant sexual dimorphism which also varies depending on cancer etiology (DOI: 10.20517/2394-5079.2020.89), it should be discussed that some endocrine disorders (but not others) potentially carry the risk of hepatocarcinogenesis (J Hepatol. 2020 Jun;72(6):1159-1169; Figure 3 in: Loria, P. et al. Nat. Rev. Gastroenterol. Hepatol. 6, 236–247 (2009).
6. Conclusion
It would be advisable to discuss in this section whetjer or not (and how) an improved understanding of endocrine pathophysiology may contribute to a better definition of NAFLD heterogeneity and personalized medicine approaches in NAFLD arena (Semin Liver Dis. 2021 Nov;41(4):421-434. Adv Ther. 2021 May;38(5):2130-2158. )
MINOR
Throught the manuscript rephrase "non-alcoholic" (used in th etitle) to "nonalcoholic" (used in the main body of the manuscript): although both spellings are commonly encountered, only the later is consistent with those pioneering definitions belonging to history of disease ( Int J Mol Sci. 2020 Aug 16;21(16):5888).
Round 2
Reviewer 2 Report
Thank you very much for your correction!
This version is very valuable and desirable to accept.
Once more thank you much for your efforts.
Your sincerely
Reviewer 3 Report
Manuscript ID: metabolites-1630595 Revised version
Title: "Nonalcoholic Fatty Liver Disease and Endocrine Axes – A Scoping Review"
Authors: Marta Borges-Canha, Madalena Von-Hafe et al.
The authors have satisfactorily responded to my comments and suggestions. They have also improved the quality of their paper and made the necessary changes to their manuscript. The revised manuscript is an interesting paper focusing on an important scientific topic. Therefore, there are no further considerations.
Reviewer 4 Report
I would like to thank these Authors on accepting this Reviewer's suggestions: as a result of this the present submission is improved to a significant extent.